# Trusting SVM for Piecewise Linear CNNs

**Leonard Berrada[1], Andrew Zisserman[1] and M. Pawan Kumar[1,2]**
[1]Department of Engineering Science
  University of Oxford
[2]Alan Turing Institute
{lberrada,az,pawan}@robots.ox.ac.uk

## Abstract

We present a novel layerwise optimization algorithm for the learning objective of Piecewise-Linear Convolutional Neural Networks (PL-CNNs), a large class of convolutional neural networks. Specifically, PL-CNNs employ piecewise linear non-linearities such as the commonly used ReLU and max-pool, and an SVM classifier as the final layer. The key observation of our approach is that the problem corresponding to the parameter estimation of a layer can be formulated as a difference-of-convex (DC) program, which happens to be a latent structured SVM. We optimize the DC program using the concave-convex procedure, which requires us to iteratively solve a structured SVM problem. This allows to design an optimization algorithm with an optimal learning rate that does not require any tuning. Using the MNIST, CIFAR and ImageNet data sets, we show that our approach always improves over the state of the art variants of backpropagation and scales to large data and large network settings.

## 1 Introduction

The backpropagation algorithm is commonly employed to estimate the parameters of a convolutional neural network (CNN) using a supervised training data set (Rumelhart et al., 1986). Part of the appeal of backpropagation comes from the fact that it is applicable to a wide variety of networks, namely those that have (sub-)differentiable non-linearities and employ a (sub-)differentiable learning objective. However, the generality of backpropagation comes at the cost of a high sensitivity to its hyperparameters such as the learning rate and momentum. Standard line-search algorithms cannot be used on the primal objective function in this setting, as (i) there may not exist a step-size guaranteeing a monotonic decrease because of the use of sub-gradients, and (ii) even in the smooth case, each function evaluation requires a forward pass over the entire data set without any update, making the approach computationally unfeasible. Choosing the learning rate thus remains an open issue, with the state-of-the-art algorithms suggesting adaptive learning rates (Duchi et al., 2011; Zeiler, 2012; Kingma & Ba, 2015). In addition, techniques such as batch normalization (Ioffe & Szegedy, 2015) and dropout (Srivastava et al., 2014) have been introduced to respectively reduce the sensitivity to the learning rate and to prevent from overfitting.

With this work, we open a different line of inquiry, namely, is it possible to design more robust optimization algorithms for special but useful classes of CNNs? To this end, we focus on the networks that are commonly used in computer vision. Specifically, we consider CNNs with convolutional and dense layers that apply a set of piecewise linear (PL) non-linear operations to obtain a discriminative representation of an input image. While this assumption may sound restrictive at first, we show that commonly used non-linear operations such as ReLU and max-pool fall under the category of PL functions. The representation obtained in this way is used to classify the image via a multi-class SVM, which forms the final layer of the network. We refer to this class of networks as PL-CNN.

We design a novel, principled algorithm to optimize the learning objective of a PL-CNN. Our algorithm is a layerwise method, that is, it iteratively updates the parameters of one layer while keeping the other layers fixed. For this work, we use a simple schedule over the

layers, namely, repeated passes from the output layer to the input one. However, it may be possible to further improve the accuracy and efficiency of our algorithm by designing more sophisticated scheduling strategies. The key observation of our approach is that the parameter estimation of one layer of PL-CNN can be formulated as a difference-of-convex (DC) program that can be viewed as a latent structured SVM problem (Yu & Joachims, 2009). This allows us to solve the DC program using the concave-convex procedure (CCCP) (Yuille & Rangarajan, 2002). Each iteration of CCCP requires us to solve a convex structured SVM problem. To this end, we use the powerful block-coordinate Frank-Wolfe (BCFW) algorithm (Lacoste-Julien et al., 2013), which solves the dual of the convex program iteratively by computing the conditional gradients corresponding to a subset of training samples. In order to further improve BCFW for PL-CNNs, we extend it in three important ways. First, we introduce a trust-region term that allows us to initialize the BCFW algorithm using the current estimate of the layer parameters. Second, we reduce the memory requirement of BCFW by an order of magnitude, via an efficient representation of the feature vectors corresponding to the dense layers. Third, we show that, empirically, the number of constraints of the structural SVM problem can be reduced substantially without any loss in accuracy, which allows us to significantly reduce its time complexity.

Compared to backpropagation (Rumelhart et al., 1986) or its variants (Duchi et al., 2011; Zeiler, 2012; Kingma & Ba, 2015), our algorithm offers three advantages. First, the CCCP algorithm provides a monotonic decrease in the learning objective at each layer. Since layerwise optimization itself can be viewed as a block-coordinate method, our algorithm guarantees a monotonic decrease of the overall objective function after each layer's parameters have been updated. Second, since the dual of the SVM problem is a smooth convex quadratic program, each step of the BCFW algorithm (in the inner iteration of the CCCP) provides a monotonic increase in its dual objective. Third, since the only step-size required in our approach comes while solving the SVM dual, we can use the optimal step-size that is computed analytically during each iteration of BCFW (Lacoste-Julien et al., 2013). In other words, our algorithm has no learning rate, initial or not, that requires tuning.

Using standard network architectures and publicly available data sets, we show that our algorithm provides a boost over the state of the art variants of backpropagation for learning PL-CNNs and we demonstrate scalability of the method.

## 2   RELATED WORK

While some of the early successful approaches for the optimization of deep neural networks relied on greedy layer-wise training (Hinton et al., 2006; Bengio et al., 2007), most currently used methods are variants of backpropagation (Rumelhart et al., 1986) with adaptive learning rates, as discussed in the introduction.

At every iteration, backpropagation performs a forward pass and a backward pass on the network, and updates the parameters of each layer by stochastic or mini-batch gradient descent. This makes the choice of the learning rate critical for efficient optimization. Duchi et al. (2011) have proposed the Adagrad convex solver, which adapts the learning rate for every direction and takes into account past updates. Adagrad changes the learning rate to favor steps in gradient directions that have not been observed frequently in past updates. When applied to the non-convex CNN optimization problem, Adagrad may converge prematurely due to a rapid decrease in the learning rate (Goodfellow et al., 2016). In order to prevent this behavior, the Adadelta algorithm (Zeiler, 2012) makes the decay of the learning rate slower. It is worth noting that this fix is empirical, and to the best of our knowledge, provides no theoretical guarantees. Kingma & Ba (2015) propose a different scheme for the learning rate, called Adam, which uses an online estimation of the first and second moments of the gradients to provide centered and normalized updates. However all these methods still require the tuning of the initial learning rate to perform well.

Second-order and natural gradient optimization methods have also been a subject of attention. The focus in this line of work has been to come up with appropriate approximations to make the updates cheaper. Martens & Sutskever (2012) suggested a Hessian-free second order optimization using finite differences to approximate the Hessian and conjugate gradient to

compute the update. Martens & Grosse (2015) derive an approximation of the Fisher matrix inverse, which provides a more efficient method for natural gradient descent. Ollivier (2013) explore a set of Riemannian methods based on natural gradient descent and quasi-Newton methods to guarantee reparametrization invariance of the problem. Desjardins et al. (2015) demonstrate a scaled up natural gradient descent method by training on the ImageNet data set (Russakovsky et al., 2015). Though providing more informative updates and solid theoretical support than SGD-based approaches, these methods do not take into account the structure of the problem offered by the commonly used non-linear operations.

Our work is also related to some of the recent developments in optimization for deep learning. For example, Taylor et al. (2016) use ADMM for massive distribution of computation in a layer-wise fashion, and in particular their method will yield closed-form updates for any PL-CNN. Lee et al. (2015) propose to use targets instead of gradients to propagate information through the network, which could help to extend our algorithm. Zhang et al. (2016) derive a convex relaxation for the learning objective for a restricted class of CNNs, which also relies on solving an approximate convex problem. In (Amos et al., 2016), the authors identify convex problems for the inference task, when the neural network is a convex function of some of its inputs.

With a more theoretical approach, Goel et al. (2016) propose an algorithm to learn shallow ReLU nets with guarantees of time convergence and generalization error. Heinemann et al. (2016) show that a subclass of neural networks can be modeled as an improper kernel, which then reduces the learning problem to a simple SVM with the constructed kernel.

More generally, we believe that our hitherto unknown observation regarding the relationship between PL-CNNs and latent SVMs can (i) allow the progress made in one field to be transferred to the other and (ii) help design a new generation of principled algorithms for deep learning optimization.

## 3 Piecewise Linear Convolutional Neural Networks

A piecewise linear convolutional neural network (PL-CNN) consists of a series of convolutional layers, followed by a series of dense layers, which provides a concise representation of an input image. Each layer of the network performs two operations: a linear transformation (that is, a convolution or a matrix multiplication), followed by a piecewise linear non-linear operation such as ReLU or max-pool. The resulting representation of the image is used for classification via an SVM. In the remainder of this section, we provide a formal description of PL-CNN.

**Piecewise Linear Functions.** A piecewise linear (PL) function $f(\mathbf{u})$ is a function of the following form (Melzer, 1986):

$$f(\mathbf{u}) = \max_{i \in [m]}\{\mathbf{a}_i^\top \mathbf{u}\} - \max_{j \in [n]}\{\mathbf{b}_j^\top \mathbf{u}\}, \tag{1}$$

where $[m] = \{1, \cdots, m\}$, and $[n] = \{1, \cdots, n\}$. Each of the two maxima above is a convex function, therefore such a function $f$ is not generally convex, but it is rather a difference of two convex functions. Importantly, many commonly used non-linear operations such as ReLU or max-pool are PL functions of their input. For example, ReLU corresponds to the function $R(v) = \max\{v, 0\}$ where $v$ is a scalar. Similarly, max-pool for a $D$-dimensional vector $\mathbf{u}$ corresponds to $M(\mathbf{u}) = \max_{i \in [D]}\{\mathbf{e}_i^\top \mathbf{u}\}$, where $\mathbf{e}_i$ is a vector whose $i$-th element is 1 and all other elements are 0. Given a value of $\mathbf{u}$, we say that $(i^*, j^*)$ is the activation of the PL function at $\mathbf{u}$ if $i^* = \operatorname{argmax}_{i \in [m]}\{\mathbf{a}_i^\top \mathbf{u}\}$ and $j^* = \operatorname{argmax}_{j \in [n]}\{\mathbf{b}_j^\top \mathbf{u}\}$.

**PL-CNN Parameters.** We denote the parameters of an $L$ layer PL-CNN by $\mathcal{W} = \{W^l; l \in [L]\}$. In other words, the parameters of the $l$-th layer is defined as $W^l$. The CNN defines a composite function, that is, the output $\mathbf{z}^{l-1}$ of layer $l-1$ is the input to the layer $l$. Given the input $\mathbf{z}^{l-1}$ to layer $l$, the output is computed as $\mathbf{z}^l = \sigma^l(W^l \cdot \mathbf{z}^{l-1})$, where "·" is either a convolution or a matrix multiplication, and $\sigma^l$ is a PL non-linear function, such as ReLU or max-pool. The input to the first layer is an image $\mathbf{x}$, that is, $\mathbf{z}^0 = \mathbf{x}$. We denote

the input to the final layer by $\mathbf{z}^L = \Phi(\mathbf{x}; \mathcal{W}) \in \mathbb{R}^D$. In other words, given an image $\mathbf{x}$, the convolutional and dense layers of a PL-CNN provide a $D$-dimensional representation of $\mathbf{x}$ to the final classification layer. The final layer of a PL-CNN is a $C$ class SVM $W^{\mathrm{svm}}$, which specifies one parameter $W_y^{\mathrm{svm}} \in \mathbb{R}^D$ for each class $y \in \mathcal{Y}$.

**Prediction.**   Given an image $\mathbf{x}$, a PL-CNN predicts its class using the following rule:

$$y^* = \underset{y \in \mathcal{Y}}{\mathrm{argmax}} \, W_y^{\mathrm{svm}} \Phi(\mathbf{x}; \mathcal{W}). \tag{2}$$

In other words, the dot product of the $D$-dimensional representation of $\mathbf{x}$ with the SVM parameter for a class $y$ provides the score for the class. The desired prediction is obtained by maximizing the score over all possible classes.

**Learning Objective.**   Given a training data set $\mathcal{D} = \{(\mathbf{x}_i, y_i), i \in [N]\}$, where $\mathbf{x}_i$ is the input image and $y_i$ is its ground-truth class, we wish to estimate the parameters $\mathcal{W} \cup W^{\mathrm{svm}}$ of the PL-CNN. To this end, we minimize a regularized upper bound on the empirical risk. The risk of a prediction $y_i^*$ given the ground-truth $y_i$ is measured with a user-specified loss function $\Delta(y_i^*, y_i)$. For example, the standard $0-1$ loss has a value of 0 for a correct prediction and 1 for an incorrect prediction. Formally, the parameters of a PL-CNN are estimated using the following learning objective:

$$\min_{\mathcal{W}, W^{\mathrm{svm}}} \frac{\lambda}{2} \sum_{l \in [L] \cup \{\mathrm{svm}\}} \|W^l\|_F^2 + \frac{1}{N} \sum_{i=1}^N \max_{\bar{y}_i \in \mathcal{Y}} \left( \Delta(\bar{y}_i, y_i) + \left( W_{\bar{y}_i}^{\mathrm{svm}} - W_{y_i}^{\mathrm{svm}} \right)^T \Phi(\mathbf{x}_i; \mathcal{W}) \right). \tag{3}$$

The hyperparameter $\lambda$ denotes the relative weight of the regularization compared to the upper bound of the empirical risk. Note that, due to the presence of piecewise linear non-linearities, the representation $\Phi(\cdot; \mathcal{W})$ (and hence, the above objective) is highly non-convex in the PL-CNN parameters.

## 4   Parameter Estimation for PL-CNN

In order to enable layerwise optimization of PL-CNNs, we show that parameter estimation of a layer can be formulated as a difference-of-convex (DC) program (subsection 4.1). This allows us to use the concave-convex procedure, which solves a series of convex optimization problems (subsection 4.2). We show that each convex problem closely resembles a structured SVM objective, which can be addressed by the powerful block-coordinate Frank-Wolfe (BCFW) algorithm. We extend BCFW to improve its initialization, time complexity and memory requirements, thereby enabling its use in learning PL-CNNs (subsection 4.3). For the sake of clarity, we only provide sketches of the proofs for those propositions that are necessary for understanding the paper. The detailed proofs of the remaining propositions are provided in the Appendix.

### 4.1   Layerwise Optimization as a DC Program

Given the values of the parameters for the convolutional and the dense layers (that is, $\mathcal{W}$), the learning objective (3) is the standard SVM problem in parameters $W^{\mathrm{svm}}$. In other words, it is a convex optimization problem with several efficient solvers (Tsochantaridis et al., 2004; Joachims et al., 2009; Shalev-Shwartz et al., 2009), including the BCFW algorithm (Lacoste-Julien et al., 2013). Hence, the optimization of the final layer is a computationally easy problem. In contrast, the optimization of the parameters of a convolutional or a dense layer $l$ does not result in a convex program. In general, this problem can be arbitrarily hard to solve. However, in the case of PL-CNN, we show that the problem can be formulated as a specific type of DC program, which enables efficient optimization via the iterative use of BCFW. The key property that enables our approach is the following proposition that shows that the composition of PL functions is also a PL function.

**Proposition 1.** *Consider PL functions $g : \mathbb{R}^m \to \mathbb{R}$ and $g_i : \mathbb{R}^n \to \mathbb{R}$, for all $i \in [m]$. Define a function $f : \mathbb{R}^n \to \mathbb{R}$ as $f(\mathbf{u}) = g([g_1(\mathbf{u}), g_2(\mathbf{u}), \cdots, g_m(\mathbf{u})]^\top)$. Then $f$ is also a PL function (proof in Appendix A).*

Using the above proposition, we can reformulate the problem of optimizing the parameters of one layer of the network as a DC program. Specifically, the following proposition shows that the problem can be formulated as a latent structured SVM objective (Yu & Joachims, 2009).

**Proposition 2.** *The learning objective of a PL-CNN with respect to the parameters of the l-th layer can be specified as follows:*

$$\min_{W^l} \frac{\lambda}{2} \|W^l\|_F^2 + \frac{1}{N} \sum_{i=1}^{N} \max_{\substack{\overline{\mathbf{h}}_i \in \mathcal{H} \\ \bar{y}_i \in \mathcal{Y}}} \left( \Delta(\bar{y}_i, y_i) + (W^l)^\top \Psi(\mathbf{x}_i, \bar{y}_i, \overline{\mathbf{h}}_i) \right) - \max_{\mathbf{h}_i \in \mathcal{H}} \left( (W^l)^\top \Psi(\mathbf{x}_i, y_i, \mathbf{h}_i) \right),$$

$$(4)$$

*for an appropriate choice of the latent space $\mathcal{H}$ and joint feature vectors $\Psi(\mathbf{x}, y, \mathbf{h})$ of the input $\mathbf{x}$, the output $y$ and the latent variables $\mathbf{h}$. In other words, parameter estimation for the l-th layer corresponds to minimizing the sum of its Frobenius norm plus a PL function for each training sample.*

*Sketch of the Proof.* For a given image $\mathbf{x}$ with the ground-truth class $y$, consider the input to the layer $l$, which we denote by $\mathbf{z}^{l-1}$. Since all the layers except the $l$-th one are fixed, the input $\mathbf{z}^{l-1}$ is a constant vector, which only depends on the image $\mathbf{x}$ (that is, its value does not depend on the variables $W^l$). In other words, we can write $\mathbf{z}^{l-1} = \varphi(\mathbf{x})$.

Given the input $\mathbf{z}^{l-1}$, all the elements of the output of the $l$-th layer, denoted by $\mathbf{z}^l$, are a PL function of $W^l$ since the layer performs a linear transformation of $\mathbf{z}^{l-1}$ according to the parameters $W^l$, followed by an application of PL operations such as ReLU or max-pool. The vector $\mathbf{z}^l$ is then fed to the $(l+1)$-th layer. The output $\mathbf{z}^{l+1}$ of the $(l+1)$-th layer is a vector whose elements are PL functions of $\mathbf{z}^l$. Therefore, by proposition (1), the elements of $\mathbf{z}^{l+1}$ are a PL function of $W^l$. By applying the same argument until we reach the layer $L$, we can conclude that the representation $\Phi(\mathbf{x}; \mathcal{W})$ is a PL function of $W^l$.

Next, consider the upper bound of the empirical risk, which is specified as follows:

$$\max_{\bar{y} \in \mathcal{Y}} \left( \Delta(\bar{y}, y) + \left( W_{\bar{y}}^{\text{svm}} - W_y^{\text{svm}} \right)^T \Phi(\mathbf{x}; \mathcal{W}) \right). \tag{5}$$

Once again, since $W^{\text{svm}}$ is fixed, the above upper bound can be interpreted as a PL function of $\Phi(\mathbf{x}; \mathcal{W})$, and thus, by proposition (1), the upper bound is a PL function of $W^l$. It only remains to observe that the learning objective (3) also contains the Frobenius norm of $W^l$. Thus, it follows that the estimation of the parameters of layer $l$ can be reformulated as minimizing the sum of its Frobenius norm and the PL upper bound of the empirical risk over all training samples, as shown in problem (4). Note that we have ignored the constants corresponding to the Frobenius norm of the parameters of all the fixed layers. This constitutes an existential proof of Proposition 2. In the next paragraph, we give an intuition about the feature vectors $\Psi(\mathbf{x}_i, \bar{y}_i, \overline{\mathbf{h}}_i)$ and the latent space $\mathcal{H}$. $\square$

**Feature Vectors & Latent Space.** The exact form of the joint feature vectors depends on the explicit DC decomposition of the objective function. In Appendix B, we detail the practical computations and give an example: we construct two interleaved neural networks whose outputs define the convex and concave parts of the DC objective function. Given the explicit DC objective function, the feature vectors are given by a subgradient and can therefore be obtained by automatic differentiation.

We now give an intuition of what the latent space $\mathcal{H}$ represents. Consider an input image $\mathbf{x}$ and a corresponding latent variable $\mathbf{h} \in \mathcal{H}$. The latent variable can be viewed as a set of variables $\mathbf{h}^k, k \in \{l+1, \cdots, L\}$. In other words, each subset $\mathbf{h}^k$ of the latent variable corresponds to one of the layers of the network that follow the layer $l$. Intuitively, $\mathbf{h}^k$ represents the choice of activation at layer $k$ when going through the PL activation: for each neuron $j$ of layer $k$, $\mathbf{h}_j^k$ takes value $i$ if and only if the $i$-th piece of the piecewise linear activation is selected. For instance, $i$ is the index of the selected input in the case of a max-pooling unit.

Note that the latent space only depends on the layers that follow the current layer being optimized. This is due to the fact that the input $\mathbf{z}^{l-1}$ to the $l$-th layer is a constant vector

that does not depend on the value of $W^l$. However, the activations of all subsequent layers following the $l$-th one depend on the value of the parameters $W^l$. As a consequence, the greater the number of following layers, the greater the size of the latent space, and this growth happens to be exponential. However, as will be seen shortly, it is still possible to efficiently optimize problem (4) for all the layers of the network despite this exponential increase.

## 4.2 Concave-Convex Procedure

The optimization problem (4) is a DC program in the parameters $W^l$. This follows from the fact that the upper bound of the empirical risk is a PL function, and can therefore be expressed as the difference of two convex PL functions (Melzer, 1986). Furthermore, the Frobenius norm of $W^l$ is also a convex function of $W^l$. This observation allows us to obtain an approximate solution of problem (4) using the iterative concave-convex procedure (CCCP) (Yuille & Rangarajan, 2002).

Algorithm 1 describes the main steps of CCCP. In step 3, we impute the best value of the latent variable corresponding to the ground-truth class $y_i$ for each training sample. This imputation corresponds to the linearization step of the CCCP. The selected latent variable corresponds to a choice of activations at each non-linear layer of the network, and therefore defines a path of activations to the ground truth. Next, in step 4, we update the parameters by solving a convex optimization problem. This convex problem amounts to finding the path of activations which minimizes the maximum margin violations given the path to the ground truth defined in step 3.

The CCCP algorithm has the desirable property of providing a monotonic decrease in the objective function at each iteration. In other words, the objective function value of problem (4) at $W_t^l$ is greater than or equal to its value at $W_{t+1}^l$. Since layerwise optimization itself can be viewed as a block-coordinate algorithm for minimizing the learning objective (3), our overall algorithm provides guarantees of monotonic decrease until convergence. This is one of the main advantages of our approach compared to backpropagation and its variants, which fail to provide similar guarantees on the value of the objective function from one iteration to the next.

---

**Algorithm 1** *CCCP for parameter estimation of the l-th layer of the PL-CNN.*

---

**Require:** Data set $\mathcal{D} = \{(\mathbf{x}_i, y_i), i \in [N]\}$, fixed parameters $\{\mathcal{W} \cup W^{\mathrm{svm}}\}\backslash W^l$, initial estimate $W_0^l$.
1: t = 0
2: **repeat**
3: For each sample $(\mathbf{x}_i, y_i)$, find the best latent variable value by solving the following problem:

$$\mathbf{h}_i^* = \underset{\overline{\mathbf{h}} \in \mathcal{H}}{\mathrm{argmax}} (W_t^l)^\top \Psi(\mathbf{x}_i, y_i, \overline{\mathbf{h}}). \tag{6}$$

4: Update the parameters by solving the following convex optimization problem:

$$W_{t+1}^l = \underset{W^l}{\mathrm{argmin}} \frac{\lambda}{2}\|W^l\|_F^2 + \frac{1}{N}\sum_{i=1}^{N} \max_{\substack{\overline{y}_i \in \mathcal{Y} \\ \overline{\mathbf{h}}_i \in \mathcal{H}}} \left(\Delta(\overline{y}_i, y_i) + (W^l)^\top \Psi(\mathbf{x}_i, \overline{y}_i, \overline{\mathbf{h}}_i)\right) -$$
$$\left((W^l)^\top \Psi(\mathbf{x}_i, y_i, \mathbf{h}_i^*)\right). \tag{7}$$

5: t = t+1
6: **until** Objective function of problem (4) cannot be improved beyond a specified tolerance.

---

In order to solve the convex program (7), which corresponds to a structured SVM problem, we make use of the powerful BCFW algorithm (Lacoste-Julien et al., 2013) that solves its dual via conditional gradients. This has two main advantages: (i) as the dual is a smooth quadratic program, each iteration of BCFW provides a monotonic increase in its

objective; and (ii) the optimal step-size at each iteration can be computed analytically. This is once again in stark contrast to backpropagation, where the estimation of the step-size is still an active area of research (Duchi et al., 2011; Zeiler, 2012; Kingma & Ba, 2015). As shown by Lacoste-Julien et al. (2013), given the current estimate of the parameters $W^l$, the conditional gradient of the dual of program (7) with respect to a training sample $(\mathbf{x}_i, y_i)$ can be obtained by solving the following problem:

$$(\hat{y}_i, \hat{\mathbf{h}}_i) = \operatorname*{argmax}_{\bar{y} \in \mathcal{Y}, \overline{\mathbf{h}} \in \mathcal{H}} (W^l)^\top \Psi(\mathbf{x}_i, \bar{y}, \overline{\mathbf{h}}) + \Delta(\bar{y}, y_i). \tag{8}$$

We refer the interested reader to (Lacoste-Julien et al., 2013) for further details.

The overall efficiency of the CCCP algorithm relies on our ability to solve problems (6) and (8). At first glance, these problems may appear to be computationally intractable as the latent space $\mathcal{H}$ can be very large, especially for layers close to the input (of the order of millions of dimensions for a typical network). However, the following proposition shows that both the problems can be solved efficiently using the forward and backward passes that are employed in backpropagation.

**Proposition 3.** *Given the current estimate $W^l$ of the parameters for the l-th layer, as well as the parameter values of all the other fixed layers, problems (6) and (8) can be solved using a forward pass on the network. Furthermore, the joint feature vectors $\Psi(\mathbf{x}_i, \hat{y}_i, \hat{\mathbf{h}}_i)$ and $\Psi(\mathbf{x}_i, y_i, \mathbf{h}_i^*)$ can be computed using a backward pass on the network.*

*Sketch of the Proof.* Recall that the latent space consists of the putative activations for each PL operation in the layers following the current one. Thus, intuitively, the maximization over the latent variables corresponds to finding the exact activations of all such PL operations. In other words, we need to identify the indices of the linear pieces that are used to compute the value of the PL function in the current state of the network. For a ReLU operation, this corresponds to estimating $\max\{0, v\}$, where the input to the ReLU is a scalar $v$. Similarly, for a max-pool operation, this corresponds to estimating $\max_i\{\mathbf{e}_i^\top \mathbf{u}\}$, where $\mathbf{u}$ is the input vector to the max-pool. This is precisely the computation that the forward pass of backpropagation performs. Given the activations, the joint feature vector is the subgradient of the sample with respect to the current layer. Once again, this is precisely what is computed during the backward pass of the backpropagation algorithm. □

An example is constructed in Appendix B to illustrate how to compute the feature vectors in practice.

### 4.3 Improving the BCFW Algorithm

As the BCFW algorithm was originally designed to solve a structured SVM problem, it requires further extensions to be suitable for training a PL-CNN. In what follows, we present three such extensions that improve the initialization, memory requirements and time complexity of the BCFW algorithm respectively.

**Trust-Region for Initialization.** The original BCFW algorithm starts with an initial parameter $W^l = \mathbf{0}$ (that is, all the parameters are set to 0). The reason for this initialization is that it is possible to compute the dual variables that correspond to the $\mathbf{0}$ primal variable. However, since our algorithm visits each layer of the network several times, it would be desirable to initialize its parameters using its current value $W_l^t$. To this end, we introduce a trust-region in the constraints of problem (7), or equivalently, an $\ell_2$ norm based proximal term in its objective function (Parikh & Boyd, 2014). The following proposition shows that this has the desired effect of initializing the BCFW algorithm close to the current parameter values.

**Proposition 4.** *By adding a proximal term $\frac{\mu}{2}\|W^l - W_t^l\|_F^2$ to the objective function in (7), we can compute a feasible dual solution whose corresponding primal solution is equal to $\frac{\mu}{\lambda+\mu}W_t^l$. Furthermore, the addition of the proximal term still allows us to efficiently compute the conditional gradient using a forward-backward pass (proof in Appendix D).*

In practice, we always choose a value of $\mu = 10\lambda$: this yields an initialization of $\simeq 0.9 W_t^l$ which does not significantly change the value of the objective function.

**Efficient Representation of Joint Feature Vectors.** The BCFW algorithm requires us to store a linear combination of the feature vectors for each mini-batch. While this requirement is not too stringent for convolutional and multi-class SVM layers, where the dimensionality of the feature vectors is small, it becomes prohibitively expensive for dense layers. The following proposition prevents a blow-up in the memory requirements of BCFW.

**Proposition 5.** *When optimizing dense layer $l$, if $W^l \in \mathbb{R}^{p \times q}$, we can store a representation of the joint feature vectors $\Psi(\mathbf{x}, y, \mathbf{h})$ with vectors of size $p$ in problems (6) and (7). This is in contrast to the naïve approach that requires them to be of size $p \times q$.*

*Sketch of the Proof.* By Proposition (3), the feature vectors are subgradients of the hinge loss function, which we loosely denote by $\eta$ for this proof. Then by the chain rule: $\frac{\partial \eta}{\partial W^l} = \frac{\partial \eta}{\partial z^l} \frac{\partial z^l}{\partial W^l} = \frac{\partial \eta}{\partial z^l} \cdot \left( z^{l-1} \right)^T$. Noting that $z^{l-1} \in \mathbb{R}^q$ is a forward pass up until layer $l$ (independent of $W^l$), we can store only $\frac{\partial \eta}{\partial z^l} \in \mathbb{R}^p$ and still reconstruct the full feature vector $\frac{\partial \eta}{\partial W^l}$ by a forward pass and an outer product. □

**Reducing the Number of Constraints.** In order to reduce the amount of time required for the BCFW algorithm to converge, we use the structure of $\mathcal{H}$ to simplify problem (7) to a much simpler problem. Specifically, since $\mathcal{H}$ represents the activations of the network for a given sample, it has a natural decomposition over the layers: $\mathcal{H} = \mathcal{H}_1 \times ... \times \mathcal{H}_L$. We use this structure in the following observation.

**Observation 1.** *Problem (7) can be approximately solved by optimizing the dual problem on increasingly large search spaces. In other words, we start with constraints of $\mathcal{Y}$, followed by $Y \times \mathcal{H}_L$, then $\mathcal{Y} \times \mathcal{H}_L \times \mathcal{H}_{L-1}$ and so on. The algorithm converges when the primal-dual gap is below tolerance.*

The latent variables which are not optimized over are set to be the same as the ones selected for the ground truth. Experimentally, we observe that for convolutional layers (architectures in section 5), restricting the search space to $\mathcal{Y}$ yields a dual gap low enough to consider the problem has converged. This means that in practice for these layers, problem (7) can be solved by searching directions over the search space $\mathcal{Y}$ instead of the much larger $\mathcal{Y} \times \mathcal{H}$. The intuition is that the norm of the difference-of-convex decomposition grows with the number of activations selected differently in the convex and concave parts (see Appendix A for the decomposition of piecewise linear functions). This compels the path of activations to be the same in the convex and the concave part to avoid large margin violations, especially for convolutional layers which are followed by numerous non-linearities at the max-pooling layers.

## 5 Experiments

Our experiments are designed to assess the ability of LW-SVM (Layer-Wise SVM, our method) and the SGD baselines to optimize problem (3). To compare LW-SVM with the state-of-the-art variants of backpropagation, we look at the training and testing accuracies as well as the training objective value. Unlike dropout, which effectively learns an ensemble model, we learn a single model using each baseline optimization algorithm. All experiments are conducted on a GPU (Nvidia Titan X) and use Theano (Bergstra et al., 2010; Bastien et al., 2012). We compare LW-SVM with Adagrad, Adadelta and Adam. For all data sets, we start at a good solution provided by these solvers and fine-tune it with LW-SVM. We then check whether a longer run of the SGD solver reaches the same level of performance.

The practical use of the LW-SVM algorithm needs choices at the three following levels: how to select the layer to optimize (i), when to stop the CCCP on each layer (ii) and when to stop the convex optimization at each inner iteration of the CCCP (iii). These choices are detailed in the next paragraph.

The layer-wise schedule of LW-SVM is as follows: as long as the validation accuracy increases, we perform passes from the end of the network (SVM) to the first layer (i). At each pass, each layer is optimized with one outer iteration of the CCCP (ii). The inner iterations are stopped when the dual objective function does not increase by more than 1% over an epoch (iii). We point out that the dual objective function is cheap to compute since we are maintaining its value at all time. By contrast, to compute the exact primal objective function requires a forward pass over the data set without any update.

## 5.1 MNIST Data Set

**Data set & Architecture**  The training data set consists in 60,000 gray scale images of size $28 \times 28$ with 10 classes, which we split into 50,000 samples for training and 10,000 for validating. The images are normalized, and we do not use any data augmentation. The architecture used for this experiment is shown in Figure 1.

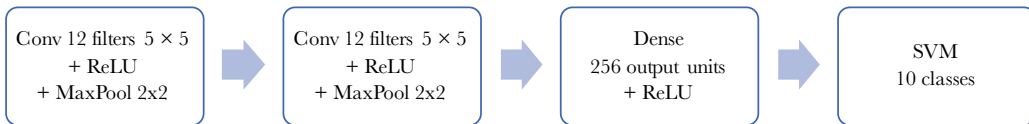

Figure 1: *Network architecture for the MNIST data set.*

**Method**  The number of epochs is set to 200, 100 and 100 for Adagrad, Adadelta and Adam - Adagrad is given more epochs as we observed it took a longer time to converge. We then use LW-SVM and compare the results on training objective, training accuracy and testing accuracy. We also let the solvers run to up to 500 epochs to verify that we have not stopped the optimization prematurely. The regularization hyperparameter $\lambda$ and the initial learning rate are chosen by cross-validation. $\lambda$ is set to 0.001 for all solvers, and the initial learning rates can be found in Appendix C. For LW-SVM, $\lambda$ is set to the same value as the baseline, and the proximal term $\mu$ to $\mu = 10\lambda = 0.01$.

Table 1: *Results on MNIST: we compare the performance of LW-SVM with SGD algorithms on three metrics: training objective, training accuracy and testing accuracy. LW-SVM outperforms Adadelta and Adam on all three metrics, with marginal improvements since those find already very good solutions.*

| Solver (epochs) | Training Objective | Training Accuracy | Time (s) | Testing Accuracy |
|---|---|---|---|---|
| Adagrad (200) | 0.027 | 99.94% | 707 | 99.22% |
| Adagrad (500) | 0.024 | 99.96% | 1759 | 99.20% |
| Adagrad (200) + LW-SVM | 0.025 | 99.94% | 707+366 | 99.21% |
| Adadelta (100) | 0.049 | 99.56% | 124 | 98.96% |
| Adadelta (500) | 0.048 | 99.48% | 619 | 99.05% |
| Adadelta (100) + LW-SVM | 0.033 | 99.85% | 124+183 | **99.24%** |
| Adam (100) | 0.038 | 99.76% | 333 | 99.19% |
| Adam (500) | 0.038 | 99.72% | 1661 | 99.23% |
| Adam (100) + LW-SVM | 0.029 | 99.89% | 333+353 | 99.23% |

**Results**  As Table 1 shows, LW-SVM systematically improves on all training objective, training accuracy and testing accuracy. In particular, it obtains the best testing accuracy when combined with Adadelta. Because each convex sub-problem is run up to sufficient convergence, the objective function of LW-SVM features of monotonic decrease at each iteration of the CCCP (blue curves in first row of Figure 2).

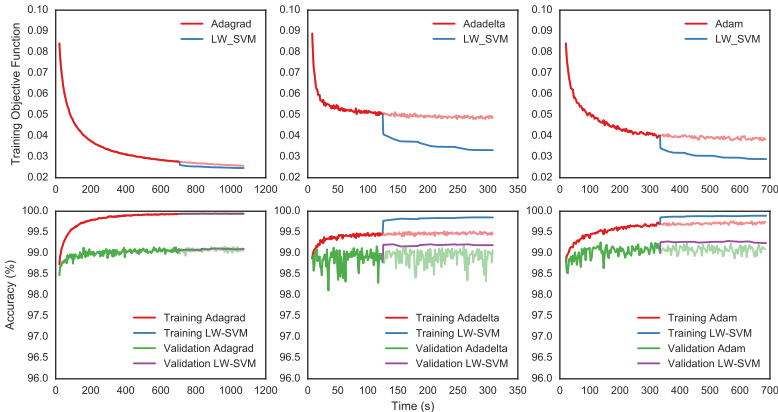

Figure 2: *Results on MNIST of Adagrad, Adadelta and Adam followed by LW-SVM. We verify that switching to LW-SVM leads to better solutions than running SGD longer (shaded continued plots).*

## 5.2 CIFAR DATA SETS

**Data sets & Architectures**   The CIFAR-10/100 data sets are comprised of 60,000 RGB natural images of size $32 \times 32$ with 10/100 classes (Krizhevsky, 2009)). We split the training set into 45,000 training samples and 5,000 validation samples in both cases. The images are centered and normalized, and we do not use any data augmentation. To obtain a strong enough baseline, we employ (i) a pre-training with a softmax and cross-entropy loss and (ii) Batch-Normalization (BN) layers before each non-linearity.

We have experimentally found out that pre-training with a softmax layer followed by a cross-entropy loss led to better behavior and results than using an SVM loss alone. The baselines are trained with batch normalization. Once they have converged, the estimated mean and standard deviation are fixed like they would be at test time. Then batch normalization becomes a linear transformation, which can be handled by the LW-SVM algorithm. This allows us to compare LW-SVM with a baseline benefiting from batch normalization. Specifically, we use the architecture shown in Figure 3:

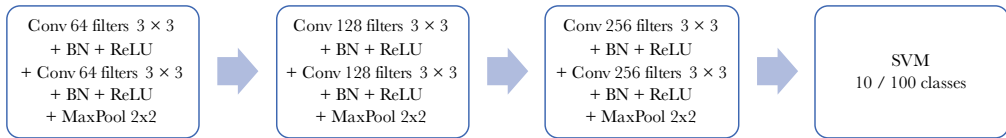

Figure 3: *Network architecture for the CIFAR data sets.*

**Method**   Again, the initial learning rates and regularization weight $\lambda$ are obtained by cross-validation, and a value of 0.001 is obtained for $\lambda$ for all solvers on both datasets. As before, $\mu$ is set to $10\lambda$. The initial learning rates are reported in Appendix C. The layer schedule and convergence criteria are as described at the beginning of the section. For each SGD optimizer, we train the network for 10 epochs with a cross-entropy loss (preceded by a softmax layer). Then it is trained with an SVM loss (without softmax) for respectively 1000, 100 and 100 epochs for Adagrad, Adadelta and Adam. This amount is doubled to verify that the baselines are not harmed by a premature stopping. Results are presented in Tables 2 and 3.

Table 2: *Results on CIFAR-10: LW-SVM outperforms Adam and Adadelta on all three metrics. It improves on Adagrad, but does not outperform it - however Adagrad takes a long time to converge and does not obtain the best generalization.*

| Solver (epochs) | Training Objective | Training Accuracy | Time (h) | Testing Accuracy |
|---|---|---|---|---|
| Adagrad (1000) | 0.059 | 98.42% | 10.58 | 83.15% |
| Adagrad (2000) | 0.009 | 100.00% | 21.14 | 83.84% |
| Adagrad (1000) + LW-SVM | 0.012 | 100.00% | 10.58+1.66 | 83.43% |
| Adadelta (100) | 0.113 | 97.96% | 0.83 | 84.42% |
| Adadelta (200) | 0.054 | 99.83% | 1.66 | 85.02% |
| Adadelta (100) + LW-SVM | 0.038 | 100.00% | 0.83+0.68 | **86.62%** |
| Adam (100) | 0.113 | 98.27% | 0.83 | 84.18% |
| Adam (200) | 0.055 | 99.76% | 1.65 | 82.55% |
| Adam (100) + LW-SVM | 0.034 | 100.00% | 0.83+1.07 | 85.52% |

Table 3: *Results on CIFAR-100: LW-SVM improves on all other solvers and obtains the best testing accuracy.*

| Solver (epochs) | Training Objective | Training Accuracy | Time (h) | Testing Accuracy |
|---|---|---|---|---|
| Adagrad (1000) | 0.201 | 95.36% | 10.68 | 54.00% |
| Adagrad (2000) | 0.044 | 99.98% | 21.20 | 54.55% |
| Adagrad (1000) + LW-SVM | 0.062 | 99.98% | 10.68+3.40 | 53.97% |
| Adadelta (100) | 0.204 | 95.68% | 0.84 | 58.71% |
| Adadelta (200) | 0.088 | 99.90% | 1.67 | 58.03% |
| Adadelta (100) + LW-SVM | 0.052 | 99.98% | 0.84+1.48 | **61.20%** |
| Adam (100) | 0.221 | 95.79% | 0.84 | 58.32% |
| Adam (200) | 0.088 | 99.87% | 1.66 | 57.81% |
| Adam (100) + LW-SVM | 0.059 | 99.98% | 0.84+1.69 | 60.17% |

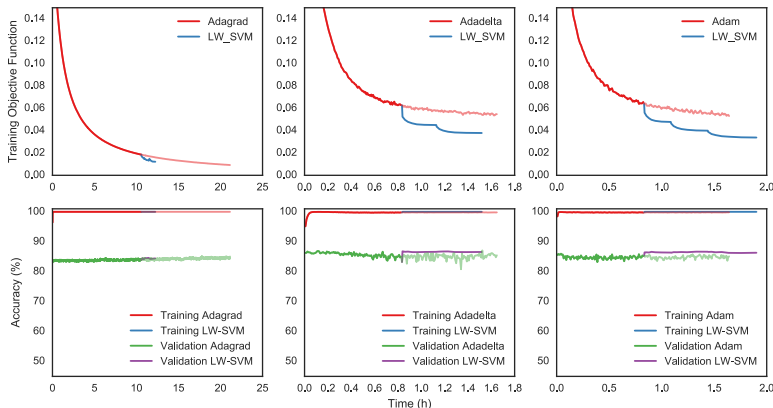

Figure 4: *Results on CIFAR-10 of Adagrad, Adadelta and Adam followed by LW-SVM. The successive drops of the training objective function with LW-SVM correspond to the passes over the layers.*

**Results** It can be seen from this set of results that LW-SVM *always* improves over the solution of the SGD algorithm, for example on CIFAR-100, decreasing the objective value of Adam from 0.22 to 0.06, or improving the test accuracy of Adadelta from 84.4% to 86.6% on

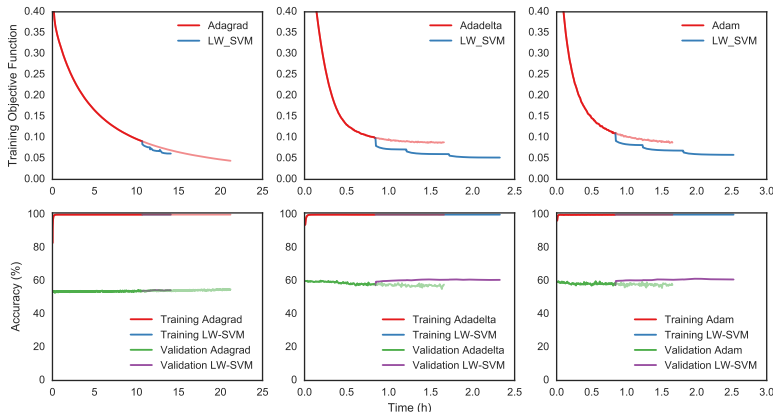

Figure 5: *Results on CIFAR-100 of Adagrad, Adadelta and Adam followed by LW-SVM. Although Adagrad keeps improving the training objective function, it takes much longer to converge and the improvement on the training and testing accuracies rapidly become marginal.*

CIFAR-10. The automatic step-size allows for a precise fine-tuning to optimize the training objective, while the regularization of the proximal term helps for better generalization.

## 5.3 IMAGENET DATA SET

We show results on the classification task of the ImageNet data set (Russakovsky et al., 2015). The ImageNet data set contains 1.2 million images for training and 50,000 images for validation, each of them mapped to one of the 1,000 classes. For this experiment we use a VGG-16 network (configuration D in (Simonyan & Zisserman, 2015)). We start with a pre-trained model as publicly available online, and we tune each of the dense layers as well as the final SVM layer with the LW-SVM algorithm. This experiment is designed to test the scalability of LW-SVM to large data sets and large networks, rather than comparing with the optimization baselines as before - indeed for any baseline, obtaining proper convergence as in previous experiments would take a very long time. We set the hyperparameters $\lambda$ to 0.001 and $\mu$ to $10\lambda$ as previously. We budget five epochs per layer, which in total takes two days of training on a single GPU (Nvidia Titan X). At training time we used centered crops of size $224 \times 224$. The evaluation method is the same as the single test scale method described in (Simonyan & Zisserman, 2015). We report the results on the validation set in Table 4, for the Pre-Trained model (PT) and the same model further optimized by LW-SVM (PT+LW-SVM):

Table 4: *Results on the 1,000-way classification challenge of ImageNet on the validation set, for the Pre-Trained model (PT) and the same model further optimized by LW-SVM (PT+LW-SVM).*

| Network | Top-1 Accuracy | Top-5 Accuracy |
|---|---|---|
| VGG-16 (PT) | 73.30% | 91.33% |
| VGG-16 (PT + LW-SVM) | 73.81% | 91.61% |

Since the objective function penalizes the top-1 error, it is logical to observe that the improvement is most important on the top-1 accuracy. Importantly, having an efficient representation of feature vectors proves to be essential for such large networks: for instance, in the optimization of the first fully connected layer with a batch-size of 100, the use of our representation lowers the memory requirements of the BCFW algorithm from 7,600GB to 20GB, which can then fit in the memory of a powerful computer.

## 6    DISCUSSION

We presented a novel layerwise optimization algorithm for a large and useful class of convolutional neural networks, which we term PL-CNNs. Our key observation is that the optimization of the parameters of one layer of a PL-CNN is equivalent to solving a latent structured SVM problem. As the problem is a DC program, it naturally lends itself to the iterative CCCP approach, which optimizes a convex structured SVM objective at each iteration. This allows us to leverage the advancements made in structured SVM optimization over the past decade to design a computationally feasible approach for learning PL-CNNs. Specifically, we use the BCFW algorithm and extend it to improve its initialization, memory requirements and time complexity. In particular, this allows our method to not require the tuning of any learning rate. Using the publicly available MNIST, CIFAR-10 and CIFAR-100 data sets, we show that our approach provides a boost for learning PL-CNNs over the state of the art backpropagation algorithms. Furthermore, we demonstrate scalability of the method with results on the ImageNet data set with a large network.

When the mean and standard deviation estimations of batch normalization are not fixed (unlike in our experiments with LW-SVM), batch normalization is not a piecewise linear transformation, and therefore cannot be used in conjunction with the BCFW algorithm for SVMs. However, it is difference-of-convex as it is a $\mathcal{C}^2$ function (Horst & Thoai, 1999). Incorporating a normalization scheme into our framework will be the object of future work. With our current methodology, LW-SVM algorithm can already be used on most standard architectures like VGG, Inception and ResNet-type architectures.

It is worth noting that other approaches for solving structured SVM problems, such as cutting-plane algorithms (Tsochantaridis et al., 2004; Joachims et al., 2009) and stochastic subgradient descent (Shalev-Shwartz et al., 2009), also rely on the efficiency of estimating the conditional gradient of the dual. Hence, all these methods are equally applicable to our setting. Indeed, the main strength of our approach is the establishment of a hitherto unknown connection between CNNs and latent structured SVMs. We believe that our observation will allow researchers to transfer the substantial existing knowledge of DC programs in general, and latent SVMs specifically, to produce the next generation of principled optimization algorithms for deep learning. In fact, there are already several such improvements that can be readily applied in our setting, which were not explored only due to a lack of time. This includes multi-plane variants of BCFW (Shah et al., 2015; Osokin et al., 2016), as well as generalizations of Frank-Wolfe such as partial linearization (Mohapatra et al., 2016).

### ACKNOWLEDGMENTS

This work was supported by the EPSRC AIMS CDT grant EP/L015987/1, the EPSRC Programme Grant Seebibyte EP/M013774/1 and Yougov. Many thanks to A. Desmaison, R. Bunel and D. Bouchacourt for the helpful discussions.

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

## A  PIECEWISE LINEAR FUNCTIONS

**Proof of Proposition (1)**  By the definition from (Melzer, 1986), we can write each function as the difference of two point-wise maxima of linear functions:

$$g(v) = \max_{j \in [m_+]} \{\mathbf{a}_i^\top \mathbf{v}\} - \max_{k \in [m_-]} \{\mathbf{b}_j^\top \mathbf{v}\}$$

$$\text{And } \forall i \in [n], g_i(u) = g_i^+(u) - g_i^-(u)$$

Where all the $g_i^+, g_i^-$ are linear point-wise maxima of linear functions. Then:

$$
\begin{aligned}
f(u) &= g([g_1(\mathbf{u}), \cdots, g_n(\mathbf{u})]^\top) \\
&= \max_{j \in [m_+]} \{\mathbf{a}_j^\top [g_1(\mathbf{u}), \cdots, g_n(\mathbf{u})]^\top\} - \max_{k \in [m_-]} \{\mathbf{b}_k^\top [g_1(\mathbf{u}), \cdots, g_n(\mathbf{u})]^\top\} \\
&= \max_{j \in [m_+]} \left\{ \sum_{i=1}^n a_{j,i} g_i(\mathbf{u}) \right\} - \max_{k \in [m_-]} \left\{ \sum_{i=1}^n b_{k,i} g_i(\mathbf{u}) \right\} \\
&= \max_{j \in [m_+]} \left\{ \sum_{i=1}^n a_{j,i} g_i^+(\mathbf{u}) - \sum_{i=1}^n a_{j,i} g_i^-(\mathbf{u}) \right\} - \max_{k \in [m_-]} \left\{ \sum_{i=1}^n b_{k,i} g_i^+(\mathbf{u}) - \sum_{i=1}^n b_{k,i} g_i^-(\mathbf{u}) \right\} \\
&= \max_{j \in [m_+]} \left\{ \sum_{i=1}^n a_{j,i} g_i^+(\mathbf{u}) + \sum_{j' \in [m_+] \setminus \{j\}} \sum_{i=1}^n a_{j,i} g_i^-(\mathbf{u}) \right\} - \sum_{j' \in [m_+]} \sum_{i=1}^n a_{j,i} g_i^-(\mathbf{u}) \\
&\quad - \max_{k \in [m_-]} \left\{ \sum_{i=1}^n b_{k,i} g_i^+(\mathbf{u}) + \sum_{k' \in [m_-] \setminus \{k\}} \sum_{i=1}^n b_{k,i} g_i^-(\mathbf{u}) \right\} + \sum_{k' \in [m_-]} \sum_{i=1}^n b_{k,i} g_i^-(\mathbf{u}) \\
&= \max_{j \in [m_+]} \left\{ \sum_{i=1}^n a_{j,i} g_i^+(\mathbf{u}) + \sum_{j' \in [m_+] \setminus \{j\}} \sum_{i=1}^n a_{j,i} g_i^-(\mathbf{u}) \right\} + \sum_{k' \in [m_-]} \sum_{i=1}^n b_{k,i} g_i^-(\mathbf{u}) \\
&\quad - \left( \max_{k \in [m_-]} \left\{ \sum_{i=1}^n b_{k,i} g_i^+(\mathbf{u}) + \sum_{k' \in [m_-] \setminus \{k\}} \sum_{i=1}^n b_{k,i} g_i^-(\mathbf{u}) \right\} + \sum_{j' \in [m_+]} \sum_{i=1}^n a_{j,i} g_i^-(\mathbf{u}) \right) \\
&= \max_{j \in [m_+]} \left\{ \sum_{i=1}^n a_{j,i} g_i^+(\mathbf{u}) + \sum_{j' \in [m_+] \setminus \{j\}} \sum_{i=1}^n a_{j,i} g_i^-(\mathbf{u}) + \sum_{k' \in [m_-]} \sum_{i=1}^n b_{k,i} g_i^-(\mathbf{u}) \right\} \\
&\quad - \max_{k \in [m_-]} \left\{ \sum_{i=1}^n b_{k,i} g_i^+(\mathbf{u}) + \sum_{k' \in [m_-] \setminus \{k\}} \sum_{i=1}^n b_{k,i} g_i^-(\mathbf{u}) + \sum_{j' \in [m_+]} \sum_{i=1}^n a_{j,i} g_i^-(\mathbf{u}) \right\}
\end{aligned}
$$

In each line of the last equality, we recognize a pointwise maximum of a linear combination of pointwise maxima of linear functions. This constitutes a pointwise maximum of linear functions.

This derivation also extends equation (10) to the multi-dimensional case by showing an explicit DC decomposition of the output.

## B  COMPUTING THE FEATURE VECTORS

We describe here how to compute the feature vectors in practice. To this end, we show how to construct two (intertwined) neural networks that decompose the objective function into a convex and a concave part. We call these Difference of Convex (DC) networks. Once the DC networks are defined, a standard forward and backward pass in the two networks yields the feature vectors for the convex and concave contribution to the objective function. First, we derive how to perform a DC decomposition in linear and non-linear layers, and then we construct an example of DC networks.

**DC Decomposition in a Linear Layer**  Let $W$ be the weights of a fixed linear layer. We introduce $W^+ = \frac{1}{2}(|W| + W)$ and $W^- = \frac{1}{2}(|W| - W)$. We can note that $W^+$ and $W^-$ have exclusively non-negative weights, and that $W = W^+ - W^-$. Say we have an input $u$ with the DC decomposition $(u^{\mathrm{cvx}}, u^{\mathrm{ccv}})$, that is: $u = u^{\mathrm{cvx}} - u^{\mathrm{ccv}}$, where both $u^{\mathrm{cvx}}$ and $u^{\mathrm{ccv}}$ are convex. Then we can decompose the output of the layer as:

$$W \cdot u = \underbrace{(W^+ \cdot u^{\mathrm{cvx}} + W^- \cdot u^{\mathrm{ccv}})}_{\text{convex}} - \underbrace{(W^- \cdot u^{\mathrm{cvx}} + W^+ \cdot u^{\mathrm{ccv}})}_{\text{convex}} \qquad (9)$$

**DC Decomposition in a Piecewise Linear Activation Layer**  For simplicity purposes, we consider that the non-linear layer is a point-wise maximum across $[K]$ scalar inputs, that is, for an input $(u_k)_{k \in [K]} \in \mathbb{R}^K$, the output is $\max_{k \in [K]} u_k$ (the general multi-dimensional case can be found in Appendix A). We suppose that we have a DC decomposition $(u_k^{\mathrm{cvx}}, u_k^{\mathrm{ccv}})$ for each input $k$. Then we can write the following decomposition for the output of the layer:

$$\max_{k \in [K]} u_k = \max_{k \in [K]} (u_k^{\mathrm{cvx}} - u_k^{\mathrm{ccv}})$$

$$= \underbrace{\max_{k \in [K]} \left( u_k^{\mathrm{cvx}} + \sum_{i \in [K], i \neq k} u_i^{\mathrm{ccv}} \right)}_{\text{convex}} - \underbrace{\sum_{k \in [K]} u_k^{\mathrm{ccv}}}_{\text{convex}} \qquad (10)$$

In particular, for a ReLU, we can write:

$$\max(u^{\mathrm{cvx}} - u^{\mathrm{ccv}}, 0) = \underbrace{\max(u^{\mathrm{cvx}}, u^{\mathrm{ccv}})}_{\text{convex}} - \underbrace{u^{\mathrm{ccv}}}_{\text{convex}} \qquad (11)$$

And for a Max-Pooling layer, one can easily verify that equation (10) is equivalent to:

$$MaxPool(u^{\mathrm{cvx}} - u^{\mathrm{ccv}}) = \underbrace{MaxPool(u^{\mathrm{cvx}} - u^{\mathrm{ccv}}) + SumPool(u^{\mathrm{ccv}})}_{\text{convex}} - \underbrace{SumPool(u^{\mathrm{ccv}})}_{\text{convex}} \qquad (12)$$

**An Example of DC Networks**  We use the previous observations to obtain a DC decomposition in any layer. We now take the example of the neural network used for the experiments on the MNIST data set, and we show how to construct the two neural networks when optimizing $W^1$, the weights of the first convolutional layer. First let us recall the architecture without decomposition:

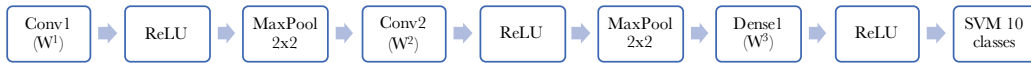

Figure 6: *Detailed network architecture for the MNIST data set.*

We want to optimize the first convolutional layer, therefore we fix all other parameters. Then we apply all operations as described in the previous paragraphs, which yields the DC networks in Figure 7.

The network graph in Figure 7 illustrates Proposition 3 for the optimization of $W^1$: suppose we are interested in $f^{\mathrm{cvx}}(\mathbf{x}, W^1)$, the convex part of the objective function for a given sample $\mathbf{x}$, and we wish to obtain the feature vector needed to perform an update of BCFW. With a forward pass, the oracle for the latent and label variables $(\hat{\mathbf{h}}, \hat{y})$ is efficiently computed; and with a backward pass, we obtain the corresponding feature vector $\Psi(\mathbf{x}, \hat{y}, \hat{\mathbf{h}})$. Indeed, we recall from problem (8) that $(\hat{\mathbf{h}}, \hat{y})$ are the latent and label variables maximizing $f^{\mathrm{cvx}}(\mathbf{x}, W^1)$. Then given $\mathbf{x}$, the forward pass in the DC networks sequentially solves the nested maximization: it maximizes the activation of the ReLU and MaxPooling units at each layer, thereby selecting the best latent variable $\hat{\mathbf{h}}$ at each non-linear layer, and maximizes the output of the SVM

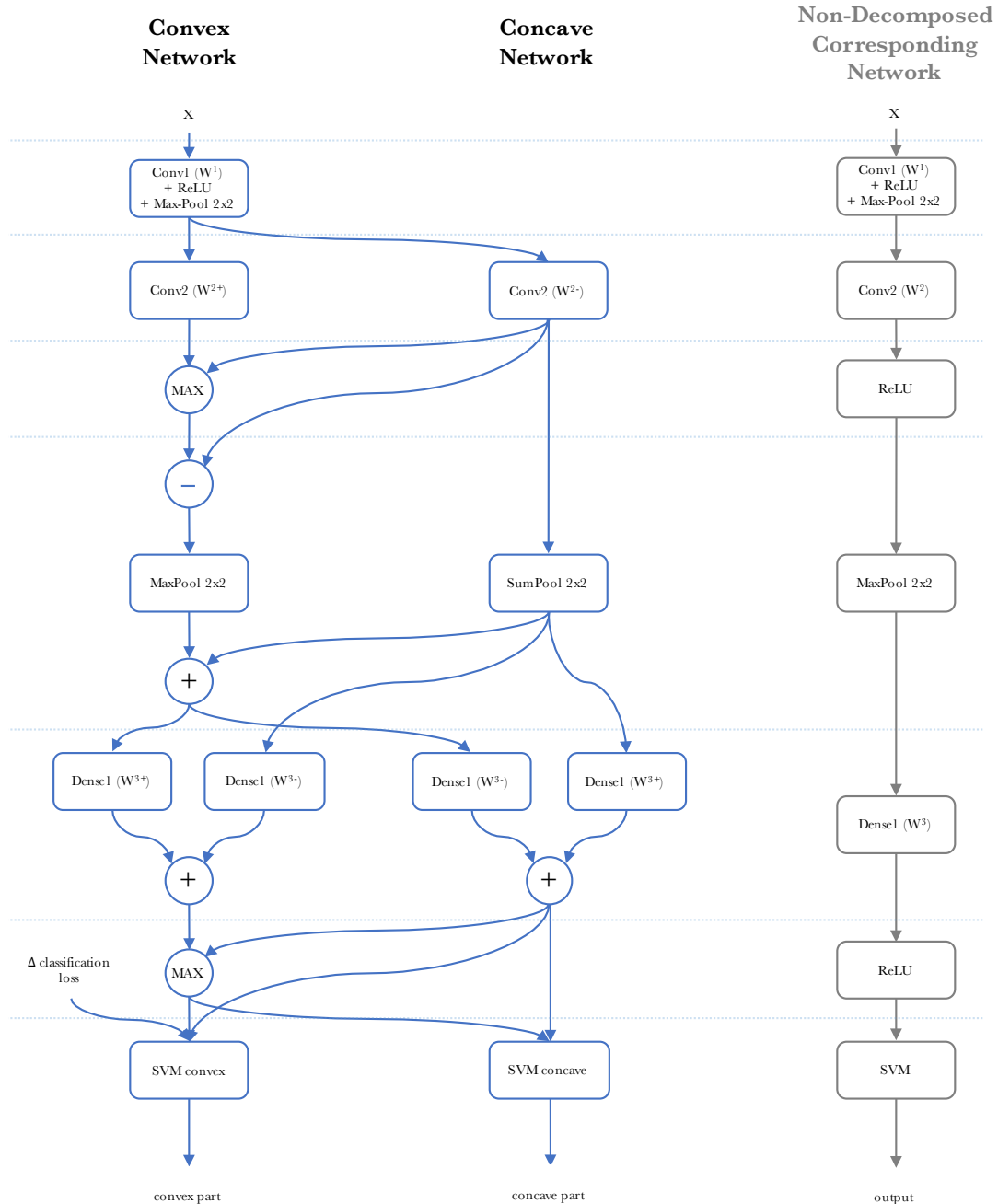

Figure 7: *Difference of Convex Networks for the optimization of Conv1 in the MNIST architecture. The two leftmost columns represent the DC networks. For each layer, the right column indicates the non-decomposed corresponding operation. Note that we represent the DC decomposition of the SVM layer as unique blocks to keep the graph simple. Given the decomposition method for linear and non-linear layers, one can write down the explicit operations without special difficulty.*

layer, thereby selecting the best label $\hat{y}$. At the end of the forward pass, $f^{\mathrm{cvx}}(\mathbf{x}, W^1)$ is therefore available as the output of the convex network, and the feature vector $\Psi(\mathbf{x}, \hat{y}, \hat{\mathbf{h}})$ can be computed as a subgradient of $f^{\mathrm{cvx}}(\mathbf{x}, W^1)$ with respect to $W^1$.

Linearizing the concave part is equivalent to fixing the activations of the DC networks, which can be done by using a fixed copy of $W^1$ at the linearization point (all other weights being fixed anyway). Then one can re-use the above reasoning to obtain the feature vectors for the linearized concave part. Altogether, this methodology allows our algorithm to be implemented in any standard deep learning library (our implementation is available at http://github.com/oval-group/pl-cnn).

## C    EXPERIMENTAL DETAILS

**Hyper-parameters**    The hyper-parameters are obtained by cross-validation with a search on powers of 10. In this section, $\eta$ will denote the initial learning rate. We denote the Softmax + Cross-Entropy loss by SCE, while SVM stands for the usual Support Vector Machines loss.

Table 5: *Hyper-parameters for the SGD solvers*

|  | MNIST | CIFAR-10 (SCE) | CIFAR-10 (SVM) | CIFAR-100 (SCE) | CIFAR-100 (SVM) |
|---|---|---|---|---|---|
| Adagrad | $\eta = 0.01$ $\lambda = 0.001$ | $\eta = 0.01$ $\lambda = 0.001$ | $\eta = 0.001$ $\lambda = 0.001$ | $\eta = 0.01$ $\lambda = 0.001$ | $\eta = 0.001$ $\lambda = 0.001$ |
| Adadelta | $\eta = 1$ $\lambda = 0.001$ | $\eta = 1$ $\lambda = 0.001$ | $\eta = 0.1$ $\lambda = 0.001$ | $\eta = 1$ $\lambda = 0.001$ | $\eta = 0.1$ $\lambda = 0.001$ |
| Adam | $\eta = 0.001$ $\lambda = 0.001$ | $\eta = 0.001$ $\lambda = 0.001$ | $\eta = 0.0001$ $\lambda = 0.001$ | $\eta = 0.001$ $\lambda = 0.001$ | $\eta = 0.0001$ $\lambda = 0.001$ |

One may note that the hyper-parameters are the same for both CIFAR-10 and CIFAR-100 for each combination of solver and loss. This makes sense since the initial learning rate mainly depends on the architecture of the network (and not so much on which particular images are fed to this network), which is very similar for the experiments on the CIFAR-10 and CIFAR-100 data sets.

## D    SVM FORMULATION & DUAL DERIVATION

**Multi-Class SVM**    Suppose we are given a data set of $N$ samples, for which every sample $i$ has a feature vector $\phi_i \in \mathbb{R}^d$ and a ground truth label $y_i \in \mathcal{Y}$. For every possible label $\bar{y}_i \in \mathcal{Y}$, we introduce the augmented feature vector $\psi_i(\bar{y}_i) \in \mathbb{R}^{|\mathcal{Y}| \times d}$ containing $\phi_i$ at index $\bar{y}_i$, $-\phi_i$ at index $y_i$, and zeros everywhere else (then $\psi_i(y_i)$ is just a vector of zeros). We also define $\Delta(\bar{y}_i, y_i)$ as the loss by choosing the output $\bar{y}_i$ instead of the ground truth $y_i$ in our task. For classification, this is the zero-one loss for example.

The SVM optimization problem is formulated as:

$$\min_{w, \xi_i} \quad \frac{\lambda}{2}\|w\|^2 + \frac{1}{N}\sum_{i=1}^{N}\xi_i$$

$$\text{subject to:} \quad \forall i \in [N], \ \forall \bar{y}_i \in \mathcal{Y}, \ \xi_i \geq w^T \psi_i(\bar{y}_i) + \Delta(y_i, \bar{y}_i)$$

Where $\lambda$ is the regularization hyperparameter. We now add a proximal term to a given starting point $w_0$:

$$\min_{w, \xi_i} \quad \frac{\lambda}{2}\|w\|^2 + \frac{\mu}{2}\|w - w_0\|^2 + \frac{1}{N}\sum_{i=1}^{N}\xi_i$$

$$\text{subject to:} \quad \forall i \in [N], \ \forall \bar{y}_i \in \mathcal{Y}, \ \xi_i \geq w^T \psi_i(\bar{y}_i) + \Delta(y_i, \bar{y}_i)$$

Factorizing the second-order polynomial in $w$, we obtain the equivalent problem (changed by a constant):

$$\min_{w,\xi_i} \quad \frac{\lambda+\mu}{2}\|w - \frac{\mu}{\lambda+\mu}w_0\|^2 + \frac{1}{N}\sum_{i=1}^{N}\xi_i$$

$$\text{subject to:} \quad \forall i \in [N], \forall \bar{y}_i \in \mathcal{Y}, \ \xi_i \geq w^T\psi_i(\bar{y}_i) + \Delta(y_i, \bar{y}_i)$$

For simplicity, we introduce the ratio $\rho = \dfrac{\mu}{\lambda+\mu}$.

**Dual Objective function**   The primal problem is:

$$\min_{w,\xi_i} \quad \frac{\lambda+\mu}{2}\|w - \rho w_0\|^2 + \frac{1}{N}\sum_{i=1}^{N}\xi_i$$

$$\text{subject to:} \quad \forall i \in [N], \forall \bar{y}_i \in \mathcal{Y}, \ \xi_i \geq w^T\psi_i(\bar{y}_i) + \Delta(y_i, \bar{y}_i)$$

The dual problem can be written as:

$$\max_{\alpha \geq 0} \min_{w,\xi_i} \quad \frac{\lambda+\mu}{2}\|w - \rho w_0\|^2 + \frac{1}{N}\sum_{i=1}^{N}\xi_i + \frac{1}{N}\sum_{i=1}^{N}\sum_{\bar{y}_i \in \mathcal{Y}} \alpha_i(\bar{y}_i)\left(\Delta(y_i, \bar{y}_i) + w^T\psi_i(\bar{y}_i) - \xi_i\right)$$

Then we obtain the following KKT conditions:

$$\forall i \in [N], \ \frac{\partial \cdot}{\partial \xi_i} = 0 \longrightarrow \sum_{\bar{y}_i \in \mathcal{Y}} \alpha_i(\bar{y}_i) = 1$$

$$\frac{\partial \cdot}{\partial w} = 0 \longrightarrow w = \rho w_0 - \underbrace{\frac{1}{N}\frac{1}{\lambda+\mu}\sum_{i=1}^{N}\sum_{\bar{y}_i \in \mathcal{Y}} \alpha_i(\bar{y}_i)\psi_i(\bar{y}_i)}_{A\alpha}$$

We also introduce $b = \frac{1}{N}(\Delta(y_i, \bar{y}_i))_{i,\bar{y}_i}$. We define $P_n(\mathcal{Y})$ as the sample-wise probability simplex:

$$u \in P_n(\mathcal{Y}) \text{ if:} \quad \forall i \in [N], \forall \bar{y}_i \in \mathcal{Y}, u_i(\bar{y}_i) \geq 0$$

$$\forall i \in [N], \ \sum_{\bar{y}_i \in \mathcal{Y}} u_i(\bar{y}_i) = 1$$

We inject back and simplify to:

$$\max_{\alpha \in P_n(\mathcal{Y})} \quad \frac{-(\lambda+\mu)}{2}\|A\alpha\|^2 + \mu w_0^T(A\alpha) + \alpha^T b$$

Finally:

$$\min_{\alpha \in P_n(\mathcal{Y})} \quad f(\alpha)$$

Where:

$$f(\alpha) \triangleq \frac{\lambda+\mu}{2}\|A\alpha\|^2 - \mu w_0^T(A\alpha) - \alpha^T b$$

**BCFW derivation**   We write $\nabla_{(i)}f$ the gradient of $f$ w.r.t. the block $(i)$ of variables in $\alpha$, padded with zeros on blocks $(j)$ for $j \neq i$. Similarly, $A_{(i)}$ and $b_{(i)}$ contain the rows of $A$ and the elements of $b$ for the block of coordinates $(i)$ and zeros elsewhere. We can write:

$$\nabla_{(i)}f(\alpha) = (\lambda+\mu)A_{(i)}^T A\alpha - \mu A_{(i)}w_0 - b_{(i)}$$

Then the search corner for the block of coordinates $(i)$ is given by:

$$s_i = \operatorname*{argmin}_{s_i'} \left( < s_i', \nabla_{(i)} f(\alpha) > \right)$$

$$= \operatorname*{argmin}_{s_i'} \left( (\lambda + \mu)\alpha^T A^T A_{(i)} s_i' - \mu w_0^T A_{(i)} s_i' - b_{(i)}^T s_i' \right)$$

We replace:

$$A\alpha = \rho w_0 - w$$

$$A_{(i)} s_i' = \frac{1}{N} \frac{1}{\lambda + \mu} \sum_{\bar{y}_i \in \mathcal{Y}} s_i'(\bar{y}_i) \psi_i(\bar{y}_i)$$

$$b_{(i)}^T s_i' = \frac{1}{N} \sum_{\bar{y}_i \in \mathcal{Y}} s_i'(\bar{y}_i) \Delta(\bar{y}_i, y_i)$$

We then obtain:

$$s_i = \operatorname*{argmin}_{s_i'} \left( -(w - \rho w_0)^T \sum_{\bar{y}_i \in \mathcal{Y}} s_i'(\bar{y}_i)\psi_i(\bar{y}_i) - w_0^T \rho \sum_{\bar{y}_i \in \mathcal{Y}} s_i'(\bar{y}_i)\psi_i(\bar{y}_i) - \sum_{\bar{y}_i \in \mathcal{Y}} s_i'(\bar{y}_i)\Delta(\bar{y}_i, y_i) \right)$$

$$= \operatorname*{argmax}_{s_i'} \left( w^T \sum_{\bar{y}_i \in \mathcal{Y}} s_i'(\bar{y}_i)\psi_i(\bar{y}_i) + \sum_{\bar{y}_i \in \mathcal{Y}} s_i'(\bar{y}_i)\Delta(\bar{y}_i, y_i) \right)$$

As expected, this maximum is obtained by setting $s_i$ to one at $y_i^* = \operatorname*{argmax}_{\bar{y}_i \in \mathcal{Y}} \left( w^T \psi_i(\bar{y}_i) + \Delta(\bar{y}_i, y_i) \right)$ and zeros elsewhere. We introduce the notation:

$$w_i = -A_{(i)} \alpha_{(i)}$$
$$l_i = b_{(i)}^T \alpha_{(i)}$$
$$w_s = -A_{(i)} s_i$$
$$l_s = b_{(i)}^T s_i$$

Then we have:

$$w_s = -\frac{1}{N} \frac{1}{\lambda + \mu} \psi(y_i^*) = -\frac{1}{N} \frac{1}{\lambda + \mu} \frac{\partial H_i(y_i^*)}{\partial w}$$

$$l_s = \frac{1}{N} \Delta(y_i, y_i^*)$$

The optimal step size in the direction of the block of coordinates $(i)$ is given by :

$$\gamma^* = \operatorname*{argmin}_{\gamma} f(\alpha + \gamma(s_i - \alpha_i))$$

The optimal step-size is given by:

$$\gamma^* = \frac{< \nabla_{(i)} f(\alpha), s_i - \alpha_i >}{(\lambda + \mu)\|A(s_i - \alpha_i)\|^2}$$

We introduce $w_d = -A\alpha = w - \rho w_0$. Then we obtain:

$$\gamma^* = \frac{(w_i - w_s)^T(w - \rho w_0) + \rho w_0^T(w_i - w_s) - \frac{1}{\lambda + \mu}(l_i - l_s)}{\|w_i - w_s\|^2}$$

$$= \frac{(w_i - w_s)^T w - \frac{1}{\lambda + \mu}(l_i - l_s)}{\|w_i - w_s\|^2}$$

And the updates are the same as in standard BCFW:

---

**Algorithm 2** *BCFW with warm start*

---

1: Let $w^{(0)} = w_0, \quad \forall i \in [N], \quad w_i^{(0)} = 0$
2: Let $l^{(0)} = 0, \quad \forall i \in [N], \quad l_i^{(0)} = 0$
3: **for** k=0...K **do**
4: Pick $i$ randomly in $\{1,..,n\}$
5: Get $y_i^* = \underset{\bar{y}_i \in \mathcal{Y}}{\operatorname{argmax}} H_i(\bar{y}_i, w^{(k)})$ and $w_s = -\frac{1}{N}\frac{1}{\lambda+\mu}\frac{\partial H_i(y_i^*, w^{(k)})}{\partial w^{(k)}}$
6: $l_s = \frac{1}{N}\Delta(y_i^*, y_i)$
7: $\gamma = \dfrac{(w_i - w_s)^T w - \frac{1}{\lambda+\mu}(l_i - l_s)}{\|w_i - w_s\|^2}$ clipped to $[0, 1]$
8: $w_i^{(k+1)} = (1-\gamma)w_i^{(k)} + \gamma w_s$
9: $l_i^{(k+1)} = (1-\gamma)l_i^{(k)} + \gamma l_s$
10: $w^{(k+1)} = w^{(k)} + w_i^{(k+1)} - w_i^{(k)} = w^{(k)} + \gamma(w_s^{(k)} - w_i^{(k)})$
11: $l^{(k+1)} = l^{(k)} + l_i^{(k+1)} - l_i^{(k)}$
12: **end for**

---

In particular, we have proved Proposition (4) in this section: $w$ is initialized to $\rho w_0$ (KKT conditions), and the direction of the conditional gradient, $w_s$, is given by $\dfrac{\partial H_i(y_i^*)}{\partial w}$, which is independent of $w_0$.

Note that the derivation of the Lagrangian dual has introduced a dual variable $\alpha_i(\bar{y}_i)$ for each linear constraint of the SVM problem (this can be replaced by $\alpha_i(\bar{\mathbf{h}}_i, (\bar{y}_i))$ if we consider latent variables). These dual variables indicate the complementary slackness not only for the output class $\bar{y}_i$, but also for each of the activation which defines a piece of the piecewise linear hinge loss. Therefore a choice of $\alpha$ defines a path of activations.

# E   Sensitivity of SGD Algorithms

Here we discuss some weaknesses of the SGD-based algorithms that we have encountered in practice for our learning objective function. These behaviors have been observed in the case of PL-CNNs, and generally may not appear in different architectures (in particular the failure to learn with high regularization goes away with the use of batch normalization layers).

## E.1   Initial Learning Rate

As mentioned in the experiments section, the choice of the initial learning rate is critical for good performance of all Adagrad, Adadelta and Adam. When the learning rate is too high, the network does not learn anything and the training and validating accuracies are stuck at random level. When it is too low, the network may take a considerably greater number of epochs to converge.

## E.2   Failures to Learn

**Regularization**   When the regularization hyper-parameter $\lambda$ is set to a value of 0.01 or higher on CIFAR-10, SGD solvers get trapped in a local minimum and fail to learn. The SGD solvers indeed fall in the local minimum of shutting down all activations on ReLUs, which provide zero-valued feature vector to the SVM loss layer (and a hinge loss of one). As a consequence, no information can be back-propagated. We plot this behavior below:

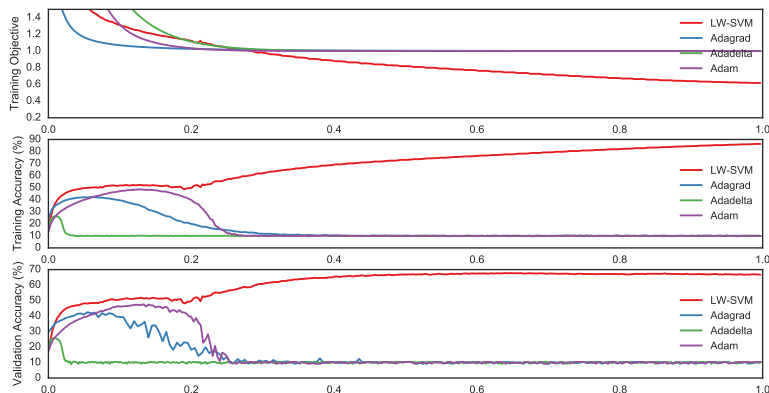

Figure 8: *Behavior of different algorithms for $\lambda = 0.01$. The x-axis has been rescaled to compare the evolution of all algorithms (real training times vary between half an hour to a few hours for the different runs).*

In this situation, the network is at a bad saddle point (note that the training and validation accuracies are stuck at random levels). Our algorithm does not fall into such bad situations, however it is not able to get out of it either: each layer is at a pathological critical point of its own objective function, which makes our algorithm unable to escape from it.

With a lower initial learning rate, the evolution is slower, but eventually the solver goes back to the bad situation presented above.

**Biases** The same failing behavior as above has been observed when not using the biases in the network. Again our algorithm is robust to this change.

