# Peer review of "Trusting SVM for Piecewise Linear CNNs"

_ICLR 2017 — accepted_

[Official Review · AnonReviewer3 · rating 6 · confidence 4 · 15 Dec 2016]
**No Title**
clarity 3

This paper proposes a new approaches for optimizing the objective of CNNs. The proposed method uses a lay-wise optimization, i.e. at each step, it optimizes the parameters in one layer of CNN while fixing the parameters in other layers. The key insight of this paper is that, for a large class of CNNs, the optimization problem at a particular can be formulated as optimizing a piecewise linear (PL) function. This PL function optimization happens to be the optimization problem commonly encountered in latent structural SVM. This connection allows this paper to borrows ideas from the latent structural SVM literature, in particular concave-convex procedure, to learn the parameters of CNNs.

Overall, the paper is well-written. Traditional, CNNs and structural SVMs are almost two separate research communties. The connection of CNNs to latent structural SVM is interesting, and might bridge the gap and facilitate the transferring of ideas between these two camps.

Of course, the proposed method also has some limitations. 1) It is limited to layer-wise optimization. Nowadays layer-wise optimization is essentially a coordinate descent algorithm and is not really a competitive strategy in learning CNNs. When you choose layer-wise optimization, you already lose something in terms of optimizing the objective (since you are using coordinate descent, instead of gradient descent). Of course, you also gain something since now you can guarantee that each coordinate descent step always improve the objective. It is not clear to me how the loss/gain balances each other. 2) This paper focues on improving the optimization of CNN objective. However, we all know that a better objective does not necessarily correspond to a good model (e.g. due to overfitting). Although the SGD with backprop in standard CNN learning does not always improve the solution of the objective (unlike the proposed method in this paper), but to me, this might be a good thing since it can prevent overfitting (the goal of learning is not to get better solution for the optimization problem in the first place -- the optimization problem is merely a proxy to learn a model with good generalization ability).

The experiment is a bit weak.
1) Only CIFAR10 is used. This is a very small dataset by today's standard, while CNNs are typically used in large-scale datasets, such as ImageNet. It is not clear whether the conclusions of this paper still hold when applied on ImageNet.

2) This paper only compares with a crippled variant of SGD (without batch normalization, dropout, etc). Although this paper mentions that the reason is that it wants to focus on optimization. But I mentioned earlier, SGD is not designed to purely obtain the best solution that optimizes the objective, the goal of SGD is to reasonably optimize the objective, while preventing overfitting. So the comparison to SGD purely in terms of the optimization is that meaningful in the first place.

[Official Review · AnonReviewer1 · rating 4 · confidence 4 · 17 Dec 2016]
**My thoughts**

A layer wise optimization for CNNs with ReLU activations and max-pooling is proposed and shown to correspond to a series of latent structured SVM problems. Using CCCP style optimization a monotonic decrease of the overall objective function can be guaranteed.

Summary:
———
I think the discussed insights are very interesting but not presented convincingly. Firstly, claims are emphasized which are often violated in practice (e.g., no convergence guarantees due to mini-batches), statements could be validated more convincingly (e.g., is monotone convergence a curse or a blessing), the experimental evaluation should be extended. In summary, I think the paper requires some more attention to form a compelling story.

Quality: I think some of the techniques could be described more carefully to better convey the intuition. At times apples are compared to oranges, e.g., back propagation is contrasted with CCCP.
Clarity: Some of the derivations and intuitions could be explained in more detail.
Originality: The suggested idea is reasonable albeit heuristics are required.
Significance: Since the experimental setup is somewhat limited according to my opinion, significance is hard to judge at this point in time.

Details:
————
1. I think the provided guarantees for the optimization procedure are certainly convenient theoretically but their practical relevance still needs to be demonstrated more convincingly, e.g., mini-batch optimization alleviates any form of monotonic decrease. Hence the emphasize in the paper is somewhat misguided according to my opinion and given he current experimental evaluation.

2. In spirit similar is work by B. Amos and J. Kolter, Input-Convex Deep Networks (

[Official Review · AnonReviewer2 · rating 5 · confidence 4 · 20 Dec 2016]
**No Title**

This paper presents a novel layer-wise optimization approach for learning CNN with piecewise linear nonlinearities.  The proposed approach trains piecewise linear CNNs layer by layer and reduces the sub-problem into latent structured SVM, which has been well-studied in the literature. In addition, the paper presents improvements of the BCFW algorithm used in the inner procedure. Overall, this paper is interesting. However, unfortunately, the experiment is not convincing. 

Pros:

- To my best knowledge, the proposed approach is novel, and the authors provide nice theoretical analysis.
- The paper is well-written and easy to follow. 

Cons:

- Although the proposed approach can be applied in general structured prediction problem, the experiments only conduct on a simple multi-class classification task. This makes this work less compelling. 
	
- The test accuracy performance on CIFAR-10 reported in the paper doesn't look right. The accuracy of the best model reported in this paper is 70.2% while existing work often reports 90+%. For example,

[Author Response · Leonard Berrada · 14 Jan 2017 (modified: 20 Jan 2017)]
**Submission Revision**

tldr: New results on ImageNet, CIFAR-100, improved results on CIFAR-10.

We thank the reviewers for their helpful feedbacks. We list here the changes made in the revisions of the paper (version 1 being the original submission read by the reviewers).

List of changes in version 2:

1) New results with batch-normalization on CIFAR-10 (new subsection 5.2)
2) Clarification of the objective of the paper and experiments (Methods paragraph in subsection 5.1)

List of changes in version 3:

1) New results on CIFAR-10: deeper architecture for a stronger baseline (subsection 5.2)
2) New results on CIFAR-100 (subsection 5.2)
3) Re-wording of the experiments section and removal of previous experiments on CIFAR-10 (with and without batch normalization) (section 5)
4) New Appendix about the computation of the feature vectors and detailed example (Appendix B).
5) Infeasibility of standard line-search in Introduction
6) New references, including suggestions from the reviewers (section 2)
7) More compact abstract
8) Inclusion of batch normalization in Discussion (section 6)
9) Minor rewording and typo fixes throughout the paper.

List of changes in version 4:
1) Added ImageNet results (subsection 5.3)

[Final Decision · Program Chairs · 06 Feb 2017]
**ICLR committee final decision**

The authors present a novel layer-wise optimization approach for learning convolutional neural networks with piecewise linear nonlinearities. The proposed approach trains piecewise linear ConvNets layer by layer, reduces the sub-problem into latent structured SVM. 
 
 Reviewers mainly expressed concerns about the experimental results, which the authors have diligently addressed in their revised versions. While the reviewers haven't updated explicitly their reviews, I believe the changes made should have been sufficient for them to do so.
 
 Thus, I recommend this paper be accepted.